# Development of culturally sensitive pain neuroscience education materials for Hausa-speaking patients with chronic spinal pain: A modified Delphi study

Naziru Bashir Mukhtar[1,2]*, Mira Meeus[1,3,4], Ceren Gursen[5], Jibril Mohammed[2], Vincent Dewitte[1], Barbara Cagnie[1]

1 Department of Rehabilitation Sciences, Ghent University, Ghent, Belgium, 2 Department of Physiotherapy, Bayero University Kano, Kano, Nigeria, 3 Pain in Motion International Research Group, Belgium, 4 Department of Rehabilitation Sciences and Physiotherapy, University of Antwerp, Antwerpen, Belgium, 5 Department of Physiotherapy and Rehabilitation, Hacettepe University, Ankara, Turkey

* nazirubashir.mukhtar@ugent.be, nbmukhtar.pth@buk.edu.ng

## Abstract

This study aimed to develop culturally sensitive pain neuroscience education (PNE) materials for Hausa speaking patients with chronic spinal pain (CSP). PNE is a program of teaching patients about pain that has gained considerable attention in research and is increasingly used during physical therapy for patients with chronic pain. It helps in decreasing pain, disability, fear-avoidance, pain catastrophization, movement restriction, and health care utilization among patients with chronic pain. However, existing PNE materials and their application are limited to few languages and cultural inclinations. Due to the variations in pain perceptions, beliefs, and related outcomes among different population groups, culture-sensitive PNE materials addressing these outcomes are warranted. A focus-group discussion comprising 4 experts was used to adapt and develop preliminary PNE materials. Thereafter, an internet-based 3-round modified Delphi-study involving 22 experts ensued. Experts' consensus/recommendations concerning the content were used in modifying the PNE materials. Consensus was predefined as ≥75% level of (dis)agreement. Eighteen experts completed the Delphi rounds. Nineteen, 18 and 18 experts participated in rounds 1, 2 and 3 respectively, representing 86%, 94% and 100% participation rate respectively. Consensus agreement was reached in every round and content of the materials, including drawings, examples, figures and metaphors were adapted following the experts' suggestions. We therefore concluded that, culture-sensitive PNE materials for Hausa speaking patients with CSP were successfully produced. The present study also provides a direction for further research whereby the effects of culturally-sensitive PNE materials can be piloted among Hausa speaking patients with CSP.

**Data Availability Statement:** Data is freely available and has been uploaded as a supporting information

**Funding:** The author(s) received no specific funding for this work.

**Competing interests:** The authors have declared that no competing interests exist.

## Introduction

An educational program for teaching patients about pain has gained considerable attention in research [1–4] and refers to different terms such as Explain Pain [5], Therapeutic Neuroscience Education [3], and Pain Neuroscience Education (PNE) [6]. PNE teaches people in pain about the biology and physiology of their pain experience, including processes such as normal biology of pain, pain modulation, pain matrix, peripheral and central sensitization, allodynia, and neuroplasticity [7, 8]. There is growing evidence for the value of PNE in decreasing pain, disability, fear-avoidance, pain catastrophization, movement restriction, and health care utilization in people struggling with pain [7, 9, 10]. Consequently, PNE is increasingly used as part of physical-therapy for patients with chronic pain in clinical settings [4, 10, 11]. Moreover, in 2015, the International Association for the Study of Pain endorsed a "call to action" which recognized an urgent need for all countries to improve access to pain management [12].

The development and use of PNE is well-established among Caucasians [9, 13–17]. However, its access in many other world languages and cultures is lacking. In the last two decades, it has been reported that variations in culture, socioeconomic status, gender issues, and literacy levels have to be considered when developing educational tools for any population [18]. Additionally, patients' beliefs are a core part of pain perception and response, as such response to pain is influenced both by patients' beliefs about it and the emotional significance attributed to it [19]. A recent systematic review revealed evidence regarding the differences in pain beliefs, pain attitudes, and coping strategies among different cultures and ethnicities [20]. Hence, a few attempts have been made to culturally adapt PNE materials. One of these attempts involved an internet-based method that required internet-access, literacy, and at least a smartphone (for access) among Brazilians [12]. Another study among Turkish-immigrants in Belgium used home education (HE) materials that required literacy to read, and included Turkish-specific pictures and metaphors [21].

There is an urgent need to develop culturally-sensitive PNE materials for different cultures and population groups worldwide in order to increase access to PNE interventions for patients with chronic pain. Furthermore, cultural sensitivity is defined by two dimensions: surface and deep structures. Surface structure involves matching intervention materials and messages to observable superficial characteristics of the target population which may involve using people, places, language, clothing, etc. that are familiar to and preferred by the target audience, whereas the deep structure encompasses the integration of cultural, social, historical, environmental, and psychological factors that influence the target health behavior in the proposed target population [22].

Neither the translation nor the culturally-sensitive version of PNE is available in any of the indigenous African languages. Africa has several indigenous languages, of which Hausa is among the most popular [23, 24]. Hausa is one of the leading African languages in terms of number of speakers. In addition, it is the unofficial lingua franca in the West-African region and studies have reported a varying number of Hausa native speakers to be between 30 to over 50 million people [25, 26].

The existing PNE materials mainly developed in Australia [5, 13, 27], Europe [14, 16, 17], and America [9] contain pictures, examples and metaphors that may not be appropriate for Hausa population. Some of the cases and scenarios do not relate to Hausa people due to differences in culture, religion, educational levels, or technological advancements. Furthermore, the conventional HE leaflets used by previous researchers [13–15, 17, 21] may not be feasible for use in Hausa population due to high levels of adult illiteracy rates, that is estimated to be around 43% [28]. The PNE materials developed for first-generation Turkish-immigrants living in Belgium [21] is the only one nearest to Hausa culture due to overlap in religion (Islam), and

to some extent with regard to clothing and gender roles. Nevertheless, these materials used a lot of pictures, metaphors and stories that are not available in a Hausa context. Nijs et al. [6] indicated that intellectual ability and health literacy of a patient should be taken into account before using a PNE program. Consequently, the lower literacy rate compared to that of Western populations is another reason to develop culturally-sensitive PNE materials for Hausa speaking patients suffering from chronic spinal pain (CSP). Finally, Hausa communities also differ from many of those where the existing PNE materials were developed in terms of religious and gender issues [29].

Consequently, it has become important that health care programs should be culturally-sensitive and not just a mere translation of the available materials [22]. The aim of the present study was to develop specific culturally-sensitive PNE Teaching Materials (TM) for Hausa women and men experiencing CSP, and general HE material by using a 2-phase sequential design of a focus group (FG) and modified Delphi-study.

## Materials and methods

### Ethical approval

The research protocol was approved by the Hospital ethics committee of Ghent University (B670201836558). Written and verbal informed consents were obtained from the literate (via mails) and non-literate (patients) experts respectively.

### Study design

A *focus group (FG) discussion* of experts followed by an internet based *3-round modified Delphi-study*.

### Procedure

First, a FG discussion was carried out to generate the preliminary materials that were used during the Delphi rounds [30–32], following a similar procedure as described by Orhan et al. [21]. Next, a 3-round modified Delphi-study was designed to gather inputs, corrections, and consensus of experts using a series of online questionnaires [21, 32, 33].

### Focus group

The preliminary PNE teaching and HE materials were developed through FG discussions. Three to 4 experts are enough for a FG meetings when participants have specialized knowledge and/or experiences to discuss [34, 35], even though using 4 to 6 experts is becoming increasingly popular because smaller groups are easier to recruit, host and are more comfortable for participants [36]. Therefore, 4 experts were included in this FG discussion (these experts were different from those in the 'Delphi expert panel'). Since there is no limit to the number of meetings in FG, as it can vary from 1 to several meetings based on the researchers' need and saturation [37], this FG had 3 meetings and each meeting lasted for up to 1 hour [38].

Experts in FG meetings are individuals that generally have superior knowledge about the topic, but there is no clear definition for such expertise [39]. Therefore, all the 4 experts were physiotherapists who have experience in the PNE concept (with a minimum of 1 peer-reviewed publication in PNE). In addition, 1 of the experts was a native Hausa language speaker. The first author (NBM) moderated the meetings and recorded all discussions. The content of the existing PNE teaching and HE materials [5, 21] was discussed during the FG meetings. The main message of 'Explain Pain' [5] was preserved. This content comprises the importance of pain, differences between acute and chronic pain, how pain originates in the

nervous system, what makes pain to persist for long time, and the sensitivity of the nervous system [5]. For the purpose of explaining this, contents, pictures, drawings, metaphors, and stories were also discussed in relation to Hausa culture.

The FG resulted in the preliminary PNE teaching and HE materials to use in the first round of the Delphi-study. The following major adaptations to the existing materials [21] were performed: (i) the development of HE materials in form of an oral interview between an expert and a journalist, so that Hausa speaking patients can listen to the interview, since majority of them cannot read [28]; (ii) modification of the pictures in the TM with drawings that depict African/Hausa people; (ii) due to cultural and religious peculiarities, separate PNE TM for male and female patients were developed [21]; (iv) metaphors, examples and stories that were thought to be inappropriate for Hausa speaking patients were changed or modified to fit Hausa contexts.

The teaching and HE materials were produced in 2 languages; a Hausa version for Hausa speaking experts and an English version for experts that do not speak Hausa. Firstly, all the materials were developed in the English language and then translated into Hausa language by the first author who is fluent in English and a native Hausa speaker. The translated materials, together with the English materials, were sent to 2 Hausa language experts for corrections [21]. This translation procedure was used after each round of the Delphi-rounds. All drawings in the materials were performed by a professional artist based on the feedback of the experts.

Throughout the manuscript, *TM* refer to the PowerPoint slides developed for teaching patients about pain, whereas the *HE material* applies to the written script of the prospective oral interview that patients will listen to at home to supplement what they have learned from the TM.

**Delphi-study.** The 3-rounds Delphi ran from May 2018 to November 2018, and were conducted according to the recommendation of guidance on conducting and reporting of Delphi-studies (CREDES) [40].

**Delphi-experts.** There is no existing guideline on who an expert is, and how many experts should be recruited in a Delphi-study [41], but in this study, 4 key areas of expertise were defined; (1) PNE; (2) Hausa culture; (3) management of Hausa speaking patients with CSP; and (4) Hausa speaking patients with CSP. Therefore, the recruited experts were a combination of: (i) physiotherapists with some experience in PNE (1 published peer-reviewed paper on PNE) or Hausa culture (with one published peer-reviewed paper on Hausa cultural adaptation) or managing Hausa speaking patients with CSP (with at least 5-years of clinical experience); and (ii) Hausa speaking patients with CSP (with either neck or back pain that lasted for at least the previous 3 months). A total of 28 experts (with 7 experts from each of the 4 key expertise areas) were purposively sampled [32, 33] and invited to participate in the Delphi-study prior to round 1. Experts were identified and selected based on the network and personal contacts of the authors (NBM, MM, JM, BC).

## Round 1

Two weeks before the start of the first-round, the participants were sent e-mails containing the PNE TM (male and female) for them to study. After these 2-weeks, the questionnaires that consists of both open and closed-ended questions in English language regarding the submitted material were sent. The open-ended questions provided the experts with freedom to give any relevant feedback, while the closed-ended questions limited their responses to only PNE contexts, since many of the participants were not PNE experts.

Experts who had limited computer literacy were guided on how to fill in the online questionnaires by one of the researchers (JM), who has experience in online surveys. The experts

and patients were asked to complete and submit the questionnaires within 2 weeks. To increase the adherence rate of the experts, a reminder via e-mail with an additional 2-week grace period was sent to those that were unable to complete the questionnaires within the initial 2 weeks.

The questionnaire for round 1 consisted of 7-items on demographics and a total of 42 (31 closed-ended and 11 open-ended) PNE materials-related questions that were adapted and modified from a previous study [21].

The questions were divided into 9 sections ("*x*") which consisted of 'acute pain', 'pain biology', 'pain modulation', 'pain matrix', 'chronic pain', 'beliefs, thoughts and behaviors', 'central sensitization', 'implications', and 'general questions'. In each of the above mentioned sections (except the 'general questions'), the following 4 multiple choice questions were asked using a 5-point Likert scale, (strongly agree–agree—don't know–disagree—strongly disagree) [42]:

i. *Do you think that these slides/pages provide relevant information about "x" (each section bears its name)*?

ii. *Do you think that the stories used to describe the"x" are feasible for the Hausa population*?

iii. *Do you think that visual information (pictures) for"x" in the TM is feasible for Hausa-patients*?

iv. *Do you think the message is clear and patients will understand*?

At the end of each section, an open-ended question was asked:

i. *If you have any suggestion(s) regarding the description of"x", please write them in the box below.*

Three final and open-ended general questions were asked:

i. *What do you think about the order and the concept? Is it understandable, logical*?

ii. *General remarks*?

iii. *Further suggestions*?

The consensus level was predefined at $\geq$75% [43], which is the minimum consensus required for a decision to be made on a particular content. When 75% or more of the experts choose to 'agree' or 'strongly agree', then such content was retained. If 75% or more of the experts choose to 'disagree' or 'strongly disagree', such content was rejected. These contents were subsequently modified based on the experts' suggestions and resubmitted in the next round. In the first round, only the 2 TM (male and female) were sent to experts in order to minimize participation fatigue and drop-outs due to bulkiness of the materials.

## Round 2

In this round, the modified TM together with the HE material were sent to all Delphi-experts that participated in round 1. In case of consensus regarding the inappropriateness of an item in the TM in round 1, such items were either modified or removed from both teaching and HE materials prior to sending them for round 2. Two different online questionnaires were sent to the experts during this round: one for the HE materials and another for the modified TM. Two weeks were given (plus two weeks grace) to the experts to complete and submit the questionnaires.

The questions for the HE materials were similar to that of the TM in round 1. Except for 'pain modulation' and 'central sensitization' sections that were not included (since these 2

sections were not included in the HE material), instead, an 'introduction' section was included. The HE material questionnaire contained a total of 35 (25 closed-ended and 10 open-ended) questions. The questionnaire for the revised TM comprised a total of 31 (23 closed-ended and 8 open-ended) questions about changes/modifications done in the following sections 'acute pain', 'pain biology', 'pain modulation' and 'beliefs, thoughts and behaviors', because experts suggested for that.

Additionally, closed-ended questions were asked concerning the male and female TM after the update:

i. *Do you think the updated material provides more relevant 'a = information, b = pictures, c = stories and metaphors' for educating Hausa-patients about pain than the round 1 material*?

ii. *Do you think the message in the updated material is clear and patients will understand*?

All the questions above were graded using the same 5-point Likert scale as in round 1 and the consensus level was maintained at ≥75%.

Three general open-ended questions concerning the updated material were added: *Is there anything;*

i. *You would like us to add to this TM*?

ii. *Specific that you would like us to modify or simplify again in this TM*?

iii. *You will further suggest*?

## Round 3

After qualitatively analyzing the responses of the experts from Round 2, questions were asked to finalize the development of the materials. Six closed-ended and 2 open-ended questions were asked to finalize this round. The questions had the usual 5-point Likert scale as in the previous rounds. The questions and responses are presented in the Results section (Table 5). This round was completed within 2 weeks.

**Closure.**   After round 3, the final teaching and HE materials were updated and developed (see S1–S3 Files for the Hausa materials and S4–S6 Files for the English versions). The finalized materials were sent to the experts along with an appreciation message for participation. The HE interview was then orally conducted and recorded (S3 File) between a professional Hausa journalist and the first author (NBM).

**Data analysis.**   Content analysis [32, 44] was used to qualitatively analyze the data of each round. The first, second and last authors independently analyzed the comments of the experts. Based on the comments and suggestions of the experts, the authors identified the topics in relation to the comments of the experts, which enabled the authors to effect the necessary modifications and corrections on the materials

## Results and discussion

Four experts who are all physiotherapists participated in the FG discussion (3 PhDs and 1 MSc holders). Although 28 experts were invited to participate in the Delphi-study, only 22 agreed. Nineteen out of 22, 18 out of 19, and all 18 experts participated in the rounds 1, 2 and 3 respectively, representing 86%, 94%, and 100% participation rate for each round. Five PNE experts, 5 CSP patients, 4 Hausa culture experts and 4 physiotherapists managing CSP patients completed round 3. The demographic characteristics of the experts in each round are presented in Table 1, while the participation flowchart can be seen in Fig 1.

**Table 1. Demographics of the experts that participated in the Delphi-study.**

| | Round 1 | Round 2 | Round 3 |
|---|---|---|---|
| | (n = 19) | (n = 18) | (n = 18) |
| Age | 37± 5.49 | 41.50±6.44 | 41.50±6.44 |
| Gender | | | |
| Male | 13 (68.4) | 11 (61.1) | 11 (61.1) |
| Female | 6 (31.6) | 7 (38.9) | 7 (38.9) |
| Country of residence | | | |
| Belgium | 2 (10.5) | 2 (11.1) | 2 (11.1) |
| The Netherlands | 1 (5.3) | 1 (5.6) | 1 (5.6) |
| Nigeria | 14 (73.7) | 14 (77.8) | 14 (77.8) |
| South Africa | 1 (5.3) | 1 (5.6) | 1 (5.6) |
| Spain | 1 (5.3) | 0(0) | 0(0) |
| Role in pain education | | | |
| Clinical purposes | 13 (68.4) | 12 (66.7) | 13 (72.2) |
| Research purposes | 6 (31.6) | 6 (33.3) | 5 (27.8) |
| Experience in pain education | | | |
| Non-existing | 4 (21.1) | 3 (16.7) | 3 (16.7) |
| Heard of it | 3 (15.8) | 1 (5.6) | 1 (5.6) |
| Familiar with it, < 1year | 2 (10.5) | 3 (16.7) | 3 (16.7) |
| Familiar with it, 1-5years | 3 (15.8) | 6 (33.3) | 5 (27.8) |
| Familiar with it, 6-10years | 3 (15.8) | 2 (11.1) | 2 (11.1) |
| Familiar with it, 11+years | 4 (21.1) | 3 (16.7) | 4 (22.2) |
| Applied/received pain education program | | | |
| Yes | 10 (52.6) | 12 (66.7) | 12 (66.7) |
| No | 9 (47.4) | 6 (33.3) | 6 (33.3) |

The data are presented as mean ± standard deviation or as absolute figures (percentages)

In all 3 rounds, the experts were predominantly males (61.1–68.4%), resident in Nigeria (73.7–77.4%), and most had a clinical role in PNE (66.7–77.2%). Among the experts, a substantial proportion had either applied or received a PNE program (52–66.7%), while 21% of them were not familiar with PNE at all.

## Round 1

Table 2 shows the responses of the experts to the closed-ended questions during round 1. For all the content of the materials, the experts reached a consensus of ≥75%, and as such, they were retained. However, the experts made suggestions for changes through the open-ended questions. The experts' responses to the open-ended questions of round 1 are presented in Table 3. Since the materials were developed for Hausa speaking patients, suggestions from experts that were not familiar with the Hausa culture, especially those that contradicted suggestions of experts familiar with Hausa culture about the feasibility/cultural context of content were not used in the modification of the materials. Most of the suggestions and changes that were made were related to drawings (change or modification to fit the Hausa culture) and the simplification of information, e.g. giving an explanation of what a spinal cord is and giving some additional content like maladaptive beliefs.

## Round 2

Table 4 presents the responses of the experts to closed-ended questions during round 2. During this round, the revised TM (male and female) and the HE material were reviewed by the

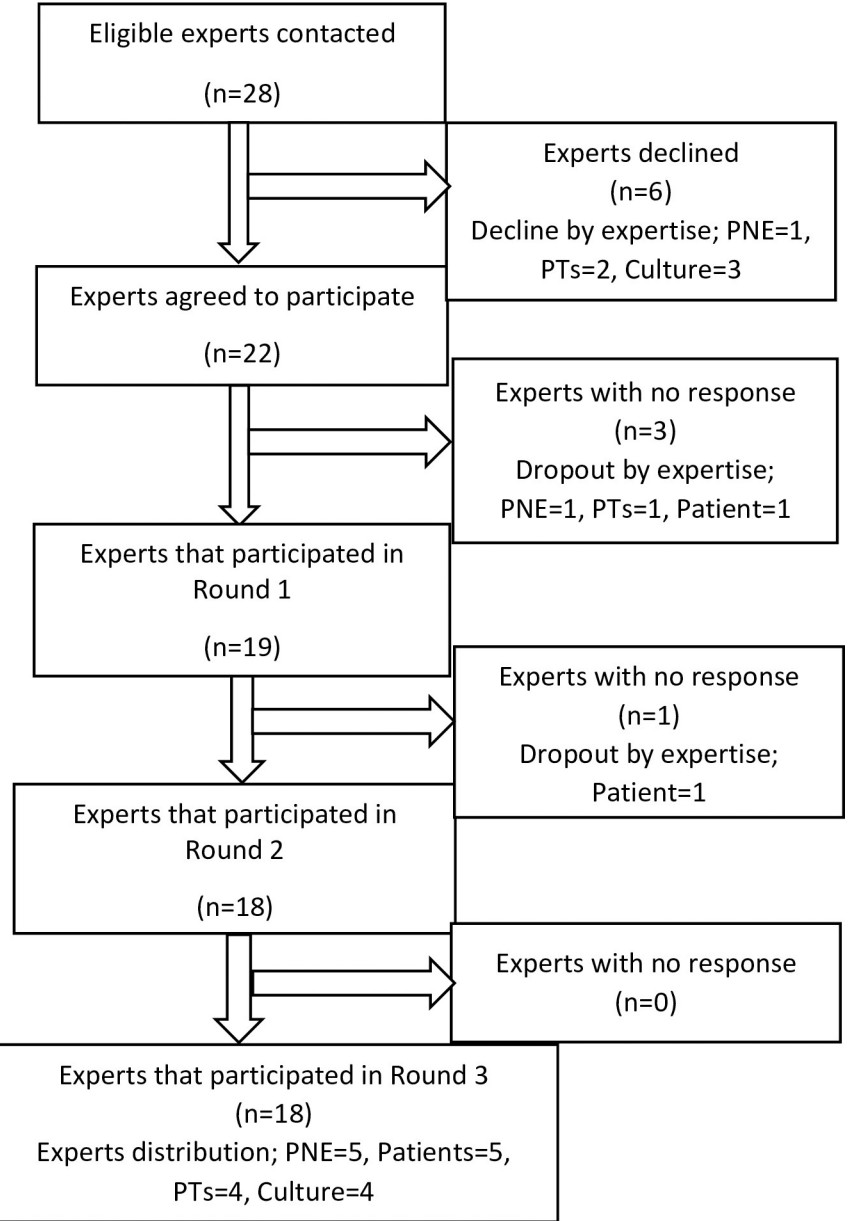

PNE=Pain Neuroscience Education, PTs=Physiotherapists

**Fig 1. Flow chart of the experts' recruitment and participation for the Delphi study.**

experts. The experts reached a consensus of ≥75% for the entire content, with some parts reaching complete consensus (100%). Therefore, additional changes in content were only done in response to the open-ended questions. Table 4 also presents the responses to the open-ended questions of experts in round 2, but suggestions requiring the use of video/animations were not considered because of resource limitations, and low technological advancements among the end-users.

**Table 2. Results of the Delphi round 1 (teaching materials for males and females) (n = 19).**

| Questions | Acute pain | Pain biology | Pain modulation | Pain matrix | Chronic pain | Beliefs, thoughts behaviors | Central sensitization | Implications |
|---|---|---|---|---|---|---|---|---|
| *Relevance of information* | | | | | | | | |
| Strongly agree/Agree | 19(100) | 18(94.7) | 18(94.7) | 17(89.5) | 16(84.2) | 17(89.5) | 17(89.5) | 18(94.7) |
| Don't know | 0(0) | 1(5.3) | 1(5.3) | 1(5.3) | 3(15.8) | 1(5.3) | 2(10.5) | 0(0) |
| Disagree/Strongly disagree | 0(0) | 0(0) | 0(0) | 1(5.3) | 0(0) | 1(5.3) | 0(0) | 1(5.3) |
| *Feasibility of the stories* | | | | | | | | |
| Strongly agree/Agree | 18(94.7) | _ | 17(89.5) | 17(89.5) | 17(89.5) | 17(89.5) | 16(84.2) | 18(94.7) |
| Don't know | 1(5.3) | _ | 2(10.5) | 1(5.3) | 2(10.5) | 2(10.5) | 2(10.5) | 0(0) |
| Disagree/Strongly disagree | 0(0) | _ | 0(0) | 1(5.3) | 0(0) | 0(0) | 1(5.3) | 1(5.3) |
| *Visual information* | | | | | | | | |
| Strongly agree/Agree | 17(89.5) | 16(84.2) | 18(94.7) | 19(100) | 18(94.7) | 18(94.7) | 17(89.5) | 18(94.7) |
| Don't know | 2(10.5) | 2(10.5) | 1(5.3) | 0(0) | 1(5.3) | 0(0) | 1(5.3) | 1(5.3) |
| Disagree/Strongly disagree | 0(0) | 1(5.3) | 0(0) | 0(0) | 0(0) | 1(5.3) | 1(5.3) | 0(0) |
| *Clarity of the message* | | | | | | | | |
| Strongly agree/Agree | 16(84.2) | 15(78.9) | 17(89.5) | 18(94.7) | 17(89.5) | 17(89.5) | 16(84.2) | 17(89.5) |
| Don't know | 1(5.3) | 2(10.5) | 2(10.5) | 1(5.3) | 1(5.3) | 2(10.5) | 1(5.3) | 2(10.5) |
| Disagree/Strongly disagree | 2(10.5) | 2(10.5) | 0(0) | 0(0) | 1(5.3) | 0(0) | 2(10.5) | 0(0) |

Data are presented as figures (percentages)

## Round 3

The results of the closed and open-ended questions in the final round are presented in Table 5. Consensus was attained in all the closed-ended question items except one (see Table 5).

The aim of the study was to develop culturally-sensitive PNE materials for Hausa speaking-patients with CSP. Preliminary teaching and HE materials were developed through a FG discussion and subsequently the final materials were developed through a 3-round Delphi-study. The TM were adapted and modified from Orhan et al. [21], who conducted a similar study among Turkish-immigrants in Belgium, whose culture is considered to be the closest to a Hausa culture among all the culturally adapted PNE materials. In this study, an audio interview was developed as HE material, given the low literacy level in that region [28]. This is supported by a previous study [11] that recommended repetitions of PNE, in different forms (verbal or other) as it helps patients to understand the theory of neurophysiology.

The PNE materials developed in this study is the first culturally-sensitive materials for an African language/culture. To our knowledge, this is also the first study that has tried to provide HE materials that can be used by the non-literates. This as necessary because the previous PNE materials were developed for populations that, as reports by UNESCO shows, have higher literacy levels than the general Hausa population [28]. Therefore, this development is in-line with the recommendations for improving access to pain management for all [12]. Although data on Hausa people being frequent listeners of audio talks is lacking, and a previous study has reported that about 89% of Nigerians to own an audio listening set [45]. This may assist the listening of the HE interview among Hausa patients.

Similarly, the development of the separate male and female TM has taken care of the importance attached to gender variations among Hausa people due to religion and culture. Varied

**Table 3. Experts' responses to open-ended questions for the Delphi round 1 (for male and female teaching materials) (n = 19).**

| Sections | Suggestions/comments of experts | Researchers' replies/actions |
|---|---|---|
| Acute pain | Change hammer injury with ankle sprain | Expert not familiar with Hausa settings |
| | Example with a room collapse not a good one | Expert not familiar with Hausa settings |
| | Give some explanation about spinal cord | Explanation given |
| | Change the metaphor of door spring- with security alarm system | Expert not familiar with Hausa settings |
| | Is example with iron applicable to- major part of the population? | Expert not familiar with Hausa settings |
| Pain biology | Change the picture on slide 16 | Change effected |
| | Modify the drawings and information of receptors | Drawing/information modified |
| | Give additional information about role of spinal cord | Additional information given |
| Pain modulation | Abdullahi is supposed to have headache-not back pain | Corrected |
| | Indicate less pain not absence of pain during shadi | Corrected |
| | Is the information below slide 19 questions- or affirmation? | Corrected to affirmations |
| | Hausa metaphor for "harm not equal to hurt" should be used | A suitable metaphor was used |
| | Upgrade pictures to be more clear | Upgraded |
| | The foot injury on slide 23 look too severe, modify the drawing | Drawing changed with a less severe injury |
| Pain matrix | Say clearly how the filters work in- Hausa traditional councils | Explanations given |
| | Give additional information regarding- roles of specific brain areas | Considered too deep for Hausa patients |
| Chronic pain | Pictures of slides 27 and 28 are not clear | Pictures upgraded |
| | Please include patients for their inputs | Patients were part of the Delphi panel |
| Beliefs, thoughts & behaviors | Give additional information about- maladaptive beliefs | Maladaptive beliefs were included |
| | Are the traditional elements important? | Expert not familiar with Hausa settings |
| | Texts below the figures are not clear | Texts made clearer |
| Central sensitization | Not sure if people will understand the- figure on slide 30 | Figure modified |
| | Simplify some descriptions because of the- low literacy among Hausas | Simple language ensured |
| | Not every patient will understand graphs | The graph was simplified with explanations |
| Implications | Information on slides 33 and 34 is not very clear | Information modified |
| General questions | Make subheadings for better understanding | Subheadings included |
| | Include more images for better understanding | More images included where necessary |

NB: Positive comments were not included in this table as they are not suggesting corrections/changes but rather recommendations. When a suggestion was made because an expert was not familiar with Hausa culture, then the suggestion was not used for modification.

gender treatment and differences exist in Hausa context, e.g. preventing girls from attending school; withdrawing girl-children from school; using girls for street hawking; and unequal treatment of children by the parent [46]. Moreover, a previous review has reported variation of pain perception, emotion and understanding between males and females among different cultures [47], as such, a uniform material for both males and females may not be appropriate. Also both teaching and HE materials bear names, examples, metaphors, and drawings/pictures that are available and familiar for Hausa speaking patients, which is in conformity with the concept of cultural sensitivity [22].

During round 1, the experts reached the minimum consensus level in almost all the closed-ended questions related to the TM. This may be a consequence of the adapted materials being modified [21] in relation to the religious inclination of most Hausa people, and to some extent their culture. The experts' responses to the open-ended questions during this round suggested the need for changes of some pictures and drawings to fit the culture and also simplification of some specific language terminologies used, which are very vital for the development of any culture sensitive tool [22].

**Table 4. Results and responses of the Delphi round 2 for the home education material and the reviewed teaching materials (n = 18).**

| Questions | Introduction | Acute pain | Pain biology | Pain modulation | Chronic pain | Beliefs, thoughts behaviors | Implications |
|---|---|---|---|---|---|---|---|
| **Home education material** | | | | | | | |
| *Relevance of information* | | | | | | | |
| Strongly agree/Agree | 18(100) | 18(100) | 18(100) | 17(94.4) | 18(100) | 18(100) | 16(88.9) |
| Don't know | 0(0) | 0(0) | 0(0) | 1(5.6) | 0(0) | 0(0) | 2(11.1) |
| Disagree/Strongly disagree | 0(0) | 0(0) | 0(0) | 0(0) | 0(0) | 0(0) | 0(0) |
| *Feasibility of the stories* | | | | | | | |
| Strongly agree/Agree | _ | _ | 16(88.9) | 17(94.4) | 17(94.4) | 17(94.4) | 16(88.9) |
| Don't know | _ | _ | 2(11.1) | 1(5.6) | 1(5.6) | 1(5.6) | 2(11.1) |
| Disagree/Strongly disagree | _ | _ | 0(0) | 0(0) | 0(0) | 0(0) | 0(0) |
| *Visual information* | | | | | | | |
| Strongly agree/Agree | _ | 16(88.8) | 17(94.4) | 17(94.4) | 16(88.9) | 17(94.4) | 17(94.4) |
| Don't know | _ | 2(11.1) | 1(5.6) | 1(5.6) | 2(11.1) | 1(5.6) | 1(5.6) |
| Disagree/Strongly disagree | _ | 0(0) | 0(0) | 0(0) | 0(0) | 0(0) | 0(0) |
| Strongly agree/Agree | 18(100) | 18(100) | 18(100) | 17(94.4) | 18(100) | 18(100) | 18(100) |
| Don't know | 0(0) | 0(0) | 0(0) | 1(5.6) | 0(0) | 0(0) | 0(0) |
| Disagree/Strongly disagree | 0(0) | 0(0) | 0(0) | 0(0) | 0(0) | 0(0) | 0(0) |

**Reviewed teaching material (male and female)**

| Questions | Acute pain | Pain biology | Pain modulation | Beliefs, thoughts behaviors | Implications | Reviewed material |
|---|---|---|---|---|---|---|
| *Relevance of information* | | | | | | |
| Strongly agree/Agree | 18(100) | 18(100) | 18(100) | 18(100) | 17(94.4) | 15(83.3) |
| Don't know | 0(0) | 0(0) | 0(0) | 0(0) | 1(5.6) | 3(16.7) |
| Disagree/Strongly disagree | 0(0) | 0(0) | 0(0) | 0(0) | 0(0) | 0(0) |
| *Feasibility of the stories* | | | | | | |
| Strongly agree/Agree | 15(83.3) | 15(83.3) | 16(88.8) | 17(94.4) | 17(94.4) | 15(83.3) |
| Don't know | 3(16.7) | 3(16.7) | 2(11.1) | 1(5.6) | 1(5.6) | 2(11.1) |
| Disagree/Strongly disagree | 0(0) | 0(0) | 0(0) | 0(0) | 0(0) | 1(5.6) |
| *Visual information* | | | | | | |
| Strongly agree/Agree | 16(88.8) | 15(83.3) | 16(88.8) | 17(94.4) | 15(83.3) | 14(77.8) |
| Don't know | 2(11.1) | 3(16.7) | 2(11.1) | 1(5.6) | 3(16.7) | 4(22.2) |
| Disagree/Strongly disagree | 0(0) | 0(0) | 0(0) | (0) | 0(0) | 0(0) |
| *Clarity of the message* | | | | | | |
| Strongly agree/Agree | 16(88.9) | 18(100) | 17(94.4) | 18(100) | 16(88.8) | 16(88.8) |
| Don't know | 2(11.1) | 0(0) | 1(5.6) | 0(0) | 2(11.1) | 1(5.6) |
| Disagree/Strongly disagree | 0(0) | 0(0) | 0(0) | 0(0) | 0(0) | 1(5.6) |

| Sections | Suggestions/comments of experts |
|---|---|
| **Home education material** | |
| Introduction | Reduce the number of names in the material |
| Acute pain | Short video clips may be of help |
| | Some responses in the interview seem too long |
| Pain biology | Illustration of normal pain biology should be simplified |
| Pain modulation | _ |
| Chronic pain | _ |

(*Continued*)

**Table 4.** (Continued)

| | |
|---|---|
| Beliefs, thoughts and behaviors | Is the example with the divorced woman a good one? |
| Implications | _ |
| General questions | Animations if possible? |
| **Reviewed teaching materials (male and female)** | |
| Acute pain | Pictures as animations to be played at sections of the hospital? |
| Pain biology | Change receptor drawings based on their functions |
| | The pictures should resemble Hausa people more |
| Pain modulation | Use few names in the stories |
| | Slide 18, make a drawing of elderlies, one with and one without pain |
| Beliefs, thoughts and behaviors | _ |
| Implications | _ |
| The whole material after review | _ |
| General questions | Video animations for watching maybe good |
| | Pictures should be more of Hausa people in outfits |

NB: Positive comments were not included in this table as they are not suggesting corrections/changes but rather recommendations. When a suggestion was made because an expert was not familiar with Hausa culture, then the suggestion was not used for modification.

The experts' consensus level increased in round 2, with 100% of experts agreeing on most of the content of the materials. This could indicate that the materials in round 2 were better accepted by the experts compared to the prior materials. Obviously, this trend is aimed for in tool development using experts' opinions [48, 49]. During round 2, there were some suggestions regarding the use of videos and animations, but such additions were not included as they may not be appropriate for Hausa speaking patients due to low technological advancement, low literacy levels, and high poverty rates among the target population [28, 50].

During the final round, the experts reached consensus on all the contents of the teaching and the HE materials and there were no suggestions that warranted an additional round. The researchers considered the TM ready for application and the HE material was then recorded in form of an oral interview by the first author with a professional Hausa journalist.

## Conclusion

It was concluded that, PNE materials that could be used to teach Hausa speaking patients with CSP and an audio interview that Hausa speaking patients can listen to at home, were successfully developed, following a well-documented, consensus building procedure. Considering the composition of the expert panel that participated in the development (i.e. physiotherapists that are experts in PNE, Hausa culture, and management of Hausa speaking patients with CSP, supplemented with the Hausa speaking patients with CSP themselves), the materials hold the promise to have high face validity and also user-friendly.

### Practice implication

The present Delphi-study may provide a direction for further research in which the effects of culturally-sensitive PNE materials can be piloted among Hausa speaking patients with CSP.

### Limitations

During the focus group discussion, only physiotherapists with PNE knowledge were involved in the development of the initial PNE materials that were subsequently used during the Delphi rounds. The lack of other professionals involved in pain management might have affected the

**Table 5. Results of the Delphi round 3 (final round) closed-ended and open-ended questions for the reviewed home education material and teaching materials (n = 18).**

| Questions | Strongly agree/ Agree | Don't know | Disagree/ Strongly disagree |
|---|---|---|---|
| **Closed-ended questions** | | | |
| A. Instead of mixing names of the characters in the teaching materials, for each of the male and female materials, we will now use one name while explaining the acute pain and the second name in explaining chronic pain up to the end of the material. Would this be better? | 18(100) | 0(0) | 0(0) |
| B. For the home education interview, would it be better to use just one male character name and one female character name (instead of two names from each gender)? | 17(94.4) | 1(5.6) | 0(0) |
| C. The drawings of the receptors are now made in different colors to differentiate their roles and sensitivity instead of being all black (refers to slides 1 and 2), do you think they are better now? | 18(100) | 0(0) | 0(0) |
| D. While presenting receptors sitting on the nerves, we have now inserted the colored receptors and their feet positions were modified with some having feet together (meaning the gate is not opened), while some have feet wide opened to allow messages to pass through (refers to slides 3 and 4). Do you think they would be better understood now? | 15(83.3) | 2(11.1) | 1(5.6) |
| E. To simplify Table 1 (refers to slide 5), we made a drawing of age-mate individuals with one in pain and the second one not in any pain (slides 6 and 7) to depict how possible it is to have similar investigation results but absence of pain. Would this slide be better understood? | 17(94.4) | 0(0) | 1(5.6) |
| F. "We can remember the jelly structure sucked by the kids when a backbone of a ram is cooked, it is the spinal cord" this statement is not necessary while explaining 'spinal cord' to Hausa patients. | 8(44.4)* | 2(11.2) | 8(44.4)* |
| **Open-ended questions** | | | |
| **Reviewed teaching materials (male and female)** | | | |
| Slide 4; the symbol besides the head is not always present and not clear | | | |
| The colors of the slides should correspond with the previous ones | | | |
| Slides 5–7; the cross in the table means pain or no pain? Better write if there is pain or no pain and explain | | | |
| Give in-text writing to explain the feet position in slides 3 and 4 | | | |
| **Reviewed home education materials** | | | |
| Example of sucking spinal cord is not necessary | | | |

* = Consensus of ≥75% was not reached and this is the last round, researchers decide to remove the statement as some experts have suggested its removal in the open-ended questions and the researchers consider its removal inconsequential

overall presentation of the PNE, however, we ensured preservation of the original PNE concept.

Another potential limitation of this study is our inability to follow a standard translation procedure for the materials developed, this is because the content of the materials was changed after each of the Delphi rounds based on experts' suggestions, and we lack resources and personnel to conduct standard translation procedures after each Delphi round. However, language experts with a minimum of PhD degrees in linguistics (Hausa language) were involved in the translation and they have been duly acknowledged.

Additionally, there was variation in the language of the experts. Therefore, the experts that did not speak Hausa language had to study the English version of the document. Also, the

experts recruited were predominantly Nigerians, and this is because Hausa people are predominantly found in Nigeria. Additionally, some of the patients recruited were not computer literate and not fluent in English language. Consequently, a research-assistant who was told not to influence their choices had to guide them on how to respond to the questionnaire.

## Supporting information

**S1 File. Pain education teaching slides (for males) in Hausa language.**
(PDF)

**S2 File. Pain education teaching slides (for females) in Hausa language.**
(PDF)

**S3 File. Home education audio interview in Hausa language.**
(MP3)

**S4 File. Pain education teaching slides (for males) in English language.**
(PDF)

**S5 File. Pain education teaching slides (for females) in English language.**
(PDF)

**S6 File. Home education audio interview (transcript) in English language.**
(PDF)

**S1 Data.**
(DOCX)

**S2 Data.**
(DOCX)

**S3 Data.**
(DOCX)

## Acknowledgments

The authors would like to thank; the experts/patients for their participation, Prof Yakubu Muhammad Azare and Dr Ahmad Shehu both of the department of Nigerian languages (Bayero University Kano, Nigeria) for their time in proofreading and reviewing the translation of the materials.

## Author Contributions

**Conceptualization:** Naziru Bashir Mukhtar, Mira Meeus, Ceren Gursen, Vincent Dewitte, Barbara Cagnie.

**Data curation:** Naziru Bashir Mukhtar, Jibril Mohammed, Barbara Cagnie.

**Formal analysis:** Naziru Bashir Mukhtar, Ceren Gursen.

**Investigation:** Naziru Bashir Mukhtar.

**Methodology:** Naziru Bashir Mukhtar, Mira Meeus, Ceren Gursen, Jibril Mohammed, Vincent Dewitte, Barbara Cagnie.

**Project administration:** Naziru Bashir Mukhtar, Jibril Mohammed.

**Resources:** Jibril Mohammed, Barbara Cagnie.

**Supervision:** Mira Meeus, Barbara Cagnie.

**Writing – original draft:** Naziru Bashir Mukhtar.

**Writing – review & editing:** Naziru Bashir Mukhtar, Mira Meeus, Ceren Gursen, Jibril Mohammed, Vincent Dewitte, Barbara Cagnie.

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
