## [Decision Letter · Decision Letter 0]

30 Dec 2020

PONE-D-20-32518

Development of culturally sensitive pain neuroscience education materials for African (Hausa) patients with chronic spinal pain: a modified Delphi study

PLOS ONE

Dear Dr. Mukhtar,

Thank you for submitting your manuscript to PLOS ONE. After careful consideration, we feel that it has some merit but does not fully meet PLOS ONE’s publication criteria as it currently stands. Therefore, we invite you to submit a revised version of the manuscript that addresses the points raised during the review process.

This is an interesting manuscript that addresses a timely topic area in physiotherapy pain eduation, particulay addressing a region of the world that is under represented in liturature. This will definately be of interest to the journal readership and I commend the authors for initiating this type of research in Africa context. That said, I will like authors to make changes to the manuscript, or write rebuttals in response to ALL the comments, querries and suggested changes from both reviwevers. 

We look forward to receiving your revised manuscript.

Kind regards,

Ukachukwu Okoroafor Abaraogu, BMR PT, MSc, PhD

Academic Editor

PLOS ONE

Journal Requirements:

2. Thank you for including your methods section "The research protocol was approved by the Hospital ethics committee of ‘X’ University."

Please amend your current ethics statement to include the full name of the ethics committee/institutional review board(s) that approved your specific study, or to remove this."

3. Thank you for stating in your methods "Informed consent was obtained from the experts." Please provide additional details regarding participant consent. In the ethics statement in the Methods and online submission information, please ensure that you have specified what type of consent you obtained (for instance, written or verbal, and if verbal, how it was documented and witnessed). Please also add all of this information to your ethics statement in the online submission form.

4. Please include additional information regarding the Delphi survey used in the study and ensure that you have provided sufficient details that others could replicate the analyses. For instance, if you developed the survey as part of this study and it is not under a copyright more restrictive than CC-BY, please include a copy, in both the original language and English, as Supporting Information.

6. We note that Supporting Information Figures in your submission contain copyrighted images. All PLOS content is published under the Creative Commons Attribution License (CC BY 4.0), which means that the manuscript, images, and Supporting Information files will be freely available online, and any third party is permitted to access, download, copy, distribute, and use these materials in any way, even commercially, with proper attribution. For more information, see our copyright guidelines: http://journals.plos.org/plosone/s/licenses-and-copyright.

6.1.    You may seek permission from the original copyright holder of Supporting Information Figures to publish the content specifically under the CC BY 4.0 license.

6.2.    If you are unable to obtain permission from the original copyright holder to publish these figures under the CC BY 4.0 license or if the copyright holder’s requirements are incompatible with the CC BY 4.0 license, please either i) remove the figure or ii) supply a replacement figure that complies with the CC BY 4.0 license. Please check copyright information on all replacement figures and update the figure caption with source information. If applicable, please specify in the figure caption text when a figure is similar but not identical to the original image and is therefore for illustrative purposes only.

7. Please include your tables as part of your main manuscript and remove the individual files. Please note that supplementary tables should be uploaded as separate "supporting information" files.

Reviewers' comments:

Reviewer's Responses to Questions

**Comments to the Author**

1. Is the manuscript technically sound, and do the data support the conclusions?

Reviewer #1: Partly

Reviewer #2: Yes

2. Has the statistical analysis been performed appropriately and rigorously? 

Reviewer #1: N/A

Reviewer #2: Yes

3. Have the authors made all data underlying the findings in their manuscript fully available?

Reviewer #1: Yes

Reviewer #2: Yes

4. Is the manuscript presented in an intelligible fashion and written in standard English?

Reviewer #1: Yes

Reviewer #2: Yes

5. Review Comments to the Author

Reviewer #1: Dear authors,

I commend you on the bold step taken by embarking on this research,it is quite apt and relevant.However,I wish to point your attention to some areas that may need some corrections and elaborations.

1.Material and method section.

a.The focus group discussion was not properly constituted.For me,I felt you needed a neurologist also as a panel member and some patients with pain to share their experience.This is the least you could have done,Physiotherapists alone may not be sufficiently resourceful enough to help with constructs for developing pain related education materials. Your content analysis could also have been highlighted in the appendix,for purpose of scrutiny.You mentioned that you adapted and developed ,I basically find these two words very confusing.Did you adapt the material from another work or did you develop a novel material for the PNG education? Please you need to be very precise and clear in your documentations.

b.You had a focus group discussion in which you developed a preliminary PNG and HE education material in English language and later translated these materials to Hausa language.For me,your translation and cultural adaptation was very substandard and shallow.Translating research materials and cultural adaptation has a standard procedure there are basically six stages which must be compulsorily adapted to make such acceptable in standard research.

STAGE 1:initial translation(Two forward translation by at least two bilingual translators they should have different profile background)

STAGE 2:Synthesis of translation

STAGE 3:Back translation

STAGE 4:Expert committee

STAGE 5:Test of the pre-final version.

STAGE 6:Submission and appraisal of all written reports by developers/committee.

All these important steps were not followed,these therefore makes your translation highly sub-standard and unacceptable for a research of these magnitude.

The processes of translation and cultural adaptation are very central in this work and less attention was paid on it but rather you spent all your time discussing the audit trail which was the Delphi study.The Delphi study is important but the main crust of this work is the translation to Hausa language.

I think you need to do a major revision of this work to make it standard and acceptable,its indeed a good and nice innovation but it must be done to international standard to make it acceptable for publication.

Reviewer #2: Reviewer's Comment:

1. Replace with Hausa Speaking Patients.

I do not see the need for Africa given that the instrument can be used by a non-African who speaks and understands Hausa very well or better than other languages.

2. "significant" on page 13: Change language except you have a statistical evidence

3. "Table 5" on page 14: Referring to the wrong table, I guess

4. "Table 6" on page 14: No table 6???

5. Insert number (n = ?) for the number of experts that participated in rounds 1 and 2 as shown in Figure 1.

6. The discussion needs to be rewritten to reflect both deductive and inductive reasoning based on other published studies.

The discussion appears like a repetition of the result section. This should be avoided.

6. PLOS authors have the option to publish the peer review history of their article (what does this mean?). If published, this will include your full peer review and any attached files.

Reviewer #1: No

Reviewer #2: **Yes: **Dr. Echezona Nelson Dominic EKECHUKWU

---

## [Author Response · Author response to Decision Letter 0]

6 Apr 2021

We wish to thank the reviewers who have provided very useful corrections and suggestions that have improved the overall quality of our work. It is our hope that the current revised version of the manuscript presents a better and more focused information that will be helpful to readers.

As requested, we have now provided a point by point response to the reviewers' comments below;

Reviewer I:

a. The focus group discussion was not properly constituted. For me, I felt you needed a neurologist also as a panel member and some patients with pain to share their experience. This is the least you could have done, Physiotherapists alone may not be sufficiently resourceful enough to help with constructs for developing pain related education materials. Your content analysis could also have been highlighted in the appendix, for purpose of scrutiny. You mentioned that you adapted and developed, I basically find these two words very confusing. Did you adapt the material from another work or did you develop a novel material for the PNG education? Please you need to be very precise and clear in your documentations.

Response: We thank the reviewer for raising these critical points. 

Firstly, the focus group discussion was aimed at providing preliminary materials that will be sent to experts during Delphi rounds, therefore, including all experts in the focus group will amount to duplicating the Delphi study. 

Secondly, in the current practice of PNE, neurologists are not considered experts unless they have specific expertise in PNE. In general, the concept of injury being synonymous to pain is always avoided in PNE, which maybe the belief of many physicians. Consequently, we considered physiotherapists with expertise in PNE to be enough for the purpose of developing a preliminary material, which was later subjected to Delphi rounds comprising experts.

Thirdly, the words adaptation and development were used in our study because, as mentioned in the method section, we adapted the pain teaching materials from an existing PNE materials (Orhan et al. cited in our method section), and we also developed a novel interview aspect that has not been in existence in the PNE field. Nevertheless, we have included some modification in the method section for better clarity.

Finally, part of the content analysis is provided for better understanding (see Table 3 and Table 4 for some of the contents analysed and the decisions taken by the authors) 

b. You had a focus group discussion in which you developed a preliminary PNG and HE education material in English language and later translated these materials to Hausa language. For me, your translation and cultural adaptation was very substandard and shallow. Translating research materials and cultural adaptation has a standard procedure there are basically six stages which must be compulsorily adapted to make such acceptable in standard research.

STAGE 1:initial translation(Two forward translation by at least two bilingual translators they should have different profile background)

STAGE 2:Synthesis of translation

STAGE 3:Back translation

STAGE 4:Expert committee

STAGE 5:Test of the pre-final version.

STAGE 6:Submission and appraisal of all written reports by developers/committee.

All these important steps were not followed, these therefore makes your translation highly sub-standard and unacceptable for a research of these magnitude.

The processes of translation and cultural adaptation are very central in this work and less attention was paid on it but rather you spent all your time discussing the audit trail which was the Delphi study. The Delphi study is important but the main crust of this work is the translation to Hausa language.

I think you need to do a major revision of this work to make it standard and acceptable, its indeed a good and nice innovation but it must be done to international standard to make it acceptable for publication.

Response: We thank the reviewer for raising this important issue. First, the aim of our study is to develop a culture-sensitive material. Therefore, the focus of our study is not on translation and cultural adaptation of an existing material. As such, a translation and cultural adaptation procedure is not reflected in our study. 

Second, the composition of the experts that participated in our study was not homogenous (English speaking and Hausa speaking). Hence, we had to first developed a preliminary material in English language through a focus group, since we lack PNE experts that are Hausa speaking. And it was these materials that were then translated through an acceptable procedure. Moreover, since the materials had to be subjected to Delphi rounds and it was expected that there will be changes based on the comments of experts after each round, translating the material via the method mentioned above may not be practical or feasible, and will not permit for the development of the tool through a Delphi procedure. Furthermore, a similar procedure has been used for the development of PNE materials in a previous study (Orhan et al -cited in our methods section). If the translation procedure suggested by the reviewer will be followed, then after each round of the Delphi we have to make similar translation which will be counterproductive and my encourage dropouts among our experts.

Reviewer II:

1. Replace with Hausa Speaking Patients.

I do not see the need for Africa given that the instrument can be used by a non-African who speaks and understands Hausa very well or better than other languages.

Response: We thank the reviewer for the valuable suggestions. Hausa speaking patients is now reflected throughout the manuscript. We have also removed African in reference to the tool we developed throughout the manuscript including the title.

2. "significant" on page 13: Change language except you have a statistical evidence

Response: The word significant has been replaced with substantial

3. "Table 5" on page 14: Referring to the wrong table, I guess

Response: We have previously collapsed Table 5 and 6 together, which earlier interfered with or Table numbering, but we have now effected the corrections.

4. "Table 6" on page 14: No table 6???

Response: Noted and corrected (As responded on item 3 above) 

5. Insert number (n = ?) for the number of experts that participated in rounds 1 and 2 as shown in Figure 1.

Response: The ‘n’ has been reflected in the Figure. We lost the ‘n’ in the earlier submission due to inappropriate box-zooming. 

6. The discussion needs to be rewritten to reflect both deductive and inductive reasoning based on other published studies.

Response: There are limited number of studies published for PNE tool development using a qualitative approach, hence we had to discuss our findings based on the existing studies including the peculiarities and novelties of our study. However, we made sure that the discussion section is improved upon, as much as possible, after also taking into account the lack of published studies in the same field adopting similar methods.

---

## [Decision Letter · Decision Letter 1]

17 May 2021

PONE-D-20-32518R1

Development of culturally sensitive pain neuroscience education materials for Hausa-speaking patients with chronic spinal pain: a modified Delphi study

PLOS ONE

Dear Dr. Mukhtar,

Thank you for submitting your manuscript to PLOS ONE. After careful consideration, we feel that it has merit but does not fully meet PLOS ONE’s publication criteria as it currently stands. Therefore, we invite you to submit a revised version of the manuscript that addresses the points raised during the review process.

In makind decision on this manuscript, I sort and recieved comments from the same reviewers who reviewed the manuscripts earlier on. Whilst reviewer 1 recomends minor correction, reviewer 2 recommend to accept. Reviewer 1 still argues that whilst having the knowledge of PNG education is an added advantage for the members of the preliminary focus group the composition of focus group for kind of study should comprise member with a broader skill including skill in the language of the instrument. Reviewer 1 will want authors to clarify the specialties of your panelist for the Delphi study. Reviewer will also want authors to acknoleged the composition of physiotherapists only panel, as well as not following the conventional delphi process as limitations to this study. Whilst I understand that certain factors might warrant adaptation of research method and procedures, I agree with the reviwer that these limitations need to be adequately acknowleged and highlighted so that readers interprete findings in the light of limitations. I will therefore invite authors to answer to reviewer 1 requeries and submit a revision. My decision is a minor revision.

Please submit your revised manuscript by 16th June 2021. If you will need more time than this to complete your revisions, please reply to this message or contact the journal office at plosone@plos.org. Please include the following items when submitting your revised manuscript:

We look forward to receiving your revised manuscript.

Kind regards,

Ukachukwu Okoroafor Abaraogu, BMR PT, MSc, PhD

Academic Editor

PLOS ONE

Journal Requirements:

Additional Editor Comments (if provided):

Dear Authors,

I have now recieved comments from the same reviewers who reviewed the manuscripts earlier on. Whilst reviewer 1 recomends minor correction, reviewer 2 recommend to accept. Reviewer 1 still argues that whilst having the knowledge of PNG education is an added advantage for the members of the preliminary focus group the composition of focus group for kind of study should comprise member with a broader skill including skill in the language of the instrument. Reviwer 1 will want authors to clarify the specialties of your panelist for the Delphi study. Reviewer will also want authors to acknoleged the composition of physiotherapists only panel, as well as not following the conventional delphi process as limitations to this study. Whilst I understand that certain factors might warrant adaptation of research method and procedures, I agree with the reviwer that these limitations need to be adequately acknowleged and highlighted so that readers interprete findings in the light of limitations. I will therefore invite authors to answer to reviewer 1 requeries and submit a revision. My decision is a minor revision.

Dr Ukachukwu Abaraogu

Department of Medical Rehabilitation University of Nigeria Nsukka/Glasgow Caledoinian University United Kigndom

Reviewers' comments:

Reviewer's Responses to Questions

**Comments to the Author**

1. If the authors have adequately addressed your comments raised in a previous round of review and you feel that this manuscript is now acceptable for publication, you may indicate that here to bypass the “Comments to the Author” section, enter your conflict of interest statement in the “Confidential to Editor” section, and submit your "Accept" recommendation.

Reviewer #1: (No Response)

Reviewer #2: All comments have been addressed

2. Is the manuscript technically sound, and do the data support the conclusions?

Reviewer #1: Partly

Reviewer #2: Yes

3. Has the statistical analysis been performed appropriately and rigorously? 

Reviewer #1: N/A

Reviewer #2: Yes

4. Have the authors made all data underlying the findings in their manuscript fully available?

Reviewer #1: Yes

Reviewer #2: Yes

5. Is the manuscript presented in an intelligible fashion and written in standard English?

Reviewer #1: Yes

Reviewer #2: Yes

6. Review Comments to the Author

Reviewer #1: 1. Having the knowledge of PNG education for me is an added advantage for the members of the preliminary focus group. That does not replace the issues of professionalism and training. By training the anesthetist is the specialist in pain management and matters, I only mentioned a neurologist in my initial comment. Preparing the preliminary material for any research is crucial and should not be trivialized. Having a panel for Delphi study does not necessary mean that they are to do the initial work of developing, their role is to modify the document. Saying that getting the right people for the preliminary document will be a waste of time is very wrong. I would rather suggest you have it as a limitation, stating that the initial document was developed only by physiotherapist with PNG education knowledge, and that no other pain specialist and professionals was involved in the initial focus group discussion. You should also note that because Orhan et al used and cited it in their work does not make the pattern a golden standard.

2. Can I ask you to kindly provide the specialties of your panelist for the Delphi study for our perusal? Pushing all responsibilities to the panelist to me may not be fair, is any of our panelist an expert in Hausa language by training? Kindly note that the six stages of translation I mentioned to you are standard and internationally accepted. I would rather suggest that you acknowledge also as a limitation your not following the process due to lack of personnel than trying to explain it away. Citing Orhan et al is not sufficient, are they experts in translation and instrumentation? Saying that your work is not focused on translation is not correct, you set out to develop a culturally sensitive pain neuroscience education material for Hausa speaking patients. The translation aspect is equally important. Goodluck.

Reviewer #2: Thanks for your corrections. My correction and comments have been addressed. The manuscript is much better now and fit for publication. Thanks.

7. PLOS authors have the option to publish the peer review history of their article (what does this mean?). If published, this will include your full peer review and any attached files.

Reviewer #1: No

Reviewer #2: No

---

## [Author Response · Author response to Decision Letter 1]

22 May 2021

May 20, 2021

Ukachukwu Okoroafor Abaraogu, BMR PT, MSc, PhD

Academic Editor

PLOSONE

Submission of re-revised manuscript (PONE-D-20-32518)

We wish to express our profound gratitude for considering our revised manuscript in the PLOS ONE Journal and recommending a minor revision. We also thank the reviewers who have gone through the revised manuscript and have provided their feedbacks that have improved the overall quality of our work. It is our hope that, after this additional revision, the current version of the manuscript presents a better and more focused information that will be helpful to readers.

As requested, we have now provided a point by point response to the Reviewer I

Reviewer I:

1. Having the knowledge of PNG education for me is an added advantage for the members of the preliminary focus group. That does not replace the issues of professionalism and training. By training the anesthetist is the specialist in pain management and matters, I only mentioned a neurologist in my initial comment. Preparing the preliminary material for any research is crucial and should not be trivialized. Having a panel for Delphi study does not necessary mean that they are to do the initial work of developing, their role is to modify the document. Saying that getting the right people for the preliminary document will be a waste of time is very wrong. I would rather suggest you have it as a limitation, stating that the initial document was developed only by physiotherapist with PNG education knowledge, and that no other pain specialist and professionals was involved in the initial focus group discussion. You should also note that because Orhan et al used and cited it in their work does not make the pattern a golden standard.

Response: We thank the reviewer for raising these critical points and we totally agree with the reviewer. We have now updated our limitation section and included the lack of other professionals in pain management as a limitation for the focus group discussion. 

2. Can I ask you to kindly provide the specialties of your panelist for the Delphi study for our perusal? Pushing all responsibilities to the panelist to me may not be fair, is any of our panelist an expert in Hausa language by training? Kindly note that the six stages of translation I mentioned to you are standard and internationally accepted. I would rather suggest that you acknowledge also as a limitation you’re not following the process due to lack of personnel than trying to explain it away. Citing Orhan et al is not sufficient, are they experts in translation and instrumentation? Saying that your work is not focused on translation is not correct, you set out to develop a culturally sensitive pain neuroscience education material for Hausa speaking patients. The translation aspect is equally important. 

Response: We thank the reviewer for raising these important issues as well. The expertise of the Delphi panel members has been presented under the ‘Delphi panel experts’ from pages 160 to 170. None of the panel experts was a Hausa language expert (by training). However, we have identified our inability to translate the developed materials using standard translation procedure as a limitation for the study (see limitations). Moreover, we have also presented the qualifications, and acknowledged the Hausa language experts involved in the translation of the materials (see limitations and acknowledgments).

---

## [Decision Letter · Decision Letter 2]

14 Jun 2021

Development of culturally sensitive pain neuroscience education materials for Hausa-speaking patients with chronic spinal pain: a modified Delphi study

PONE-D-20-32518R2

Dear Dr. Mukhtar,

We’re pleased to inform you that your manuscript has been judged scientifically suitable for publication and will be formally accepted for publication once it meets all outstanding technical requirements.

Kind regards,

Ukachukwu Okoroafor Abaraogu, BMR PT, MSc, PhD

Academic Editor

PLOS ONE

Additional Editor Comments (optional):

Reviewers' comments:

Reviewer's Responses to Questions

**Comments to the Author**

1. If the authors have adequately addressed your comments raised in a previous round of review and you feel that this manuscript is now acceptable for publication, you may indicate that here to bypass the “Comments to the Author” section, enter your conflict of interest statement in the “Confidential to Editor” section, and submit your "Accept" recommendation.

Reviewer #1: All comments have been addressed

2. Is the manuscript technically sound, and do the data support the conclusions?

Reviewer #1: Yes

3. Has the statistical analysis been performed appropriately and rigorously? 

Reviewer #1: Yes

4. Have the authors made all data underlying the findings in their manuscript fully available?

Reviewer #1: Yes

5. Is the manuscript presented in an intelligible fashion and written in standard English?

Reviewer #1: Yes

6. Review Comments to the Author

Reviewer #1: (No Response)

7. PLOS authors have the option to publish the peer review history of their article (what does this mean?). If published, this will include your full peer review and any attached files.

Reviewer #1: No

---

## [Editor Report · Acceptance letter]

18 Jun 2021

PONE-D-20-32518R2 

Development of culturally sensitive pain neuroscience education materials for Hausa-speaking patients with chronic spinal pain: a modified Delphi study 

Dear Dr. Mukhtar:

I'm pleased to inform you that your manuscript has been deemed suitable for publication in PLOS ONE. Congratulations! Your manuscript is now with our production department. 

Kind regards, 

on behalf of

Dr. Ukachukwu Okoroafor Abaraogu 

Academic Editor

PLOS ONE